# Exploring perspectives on digital smoking cessation just-in-time adaptive interventions: A focus group study with adult smokers and smoking cessation professionals

Corinna Leppin[1]*, Tosan Okpako[1], Claire Garnett[1,2], Olga Perski[1,3,4], Jamie Brown[1]

**1** Department of Behavioural Science and Health, University College London, London, United Kingdom, **2** School of Psychological Science, University of Bristol, Bristol, United Kingdom, **3** Herbert Wertheim School of Public Health and Human Longevity Science, University of California, San Diego, California, United States of America, **4** Faculty of Social Sciences, Tampere University, Tampere, Finland

* corinna.leppin.21@ucl.ac.uk

## Abstract

Technology-mediated just-in-time adaptive interventions (JITAIs), which provide users with real-time, tailored behavioural support, are a promising innovation for smoking cessation. However, a greater understanding of stakeholder, including user, perspectives on JITAIs is needed. Focus groups with UK-based adult smokers (three groups; N = 19) and smoking cessation professionals (one group; N = 5) were conducted January-June 2024. Topic guides addressed the integration of a JITAI into users' lives, preferred content and features, and data and privacy. Transcripts were analysed using inductive and deductive Framework Analysis; deductive codes were derived from the Theoretical Domains Framework and the Technology Acceptance Model. Four co-equal major themes, "Smoking Cessation Process", "JITAI Characteristics", "Perceived Value of the JITAI", and "Relationship with the JITAI", and 16 subordinate themes were identified. The smoking cessation process was described as a challenging and idiosyncratic, non-linear journey during which a JITAI should provide consistent support. Preferences for specific JITAI characteristics varied. However, participants consistently expressed that a JITAI should be highly personalised and offer both immediate, interruptive support and ambient, in-depth content. The perceived usefulness and ease of use of a JITAI were described as central to its perceived value. Participants stressed that a JITAI would need to be convenient enough to easily integrate into users' daily lives, yet disruptive enough to facilitate behaviour change. Smokers expressed that they would want their relationship with a JITAI to feel supportive and non-judgmental. They also felt a JITAI should promote their autonomy. Smoking cessation professionals stressed the importance of privacy and data protection, whereas smokers appeared more ambivalent and had mixed opinions about this topic. JITAIs need to balance aspects of competing demands in their design, such as optimising for both convenience and sufficient disruption,

**Data availability statement:** The codebook underpinning analysis of the current study is available in the Open Science Framework repository, https://osf.io/p7c9y/ (Identifier: DOI 10.17605/OSF.IO/P7C9Y).

**Funding:** This review is part of a project funded by Cancer Research UK (PRCRPG-Nov21\100002; https://www.cancerresearchuk.org/funding-for-researchers), awarded to JB. The funders had no role in study design, data collection and analysis, decision to publish, or preparation of the manuscript.

**Competing interests:** JB has received unrestricted research funding to study smoking cessation from pharmaceutical companies who manufacture smoking cessation medications (Pfizer and J&J). OP and JB act as unpaid scientific advisors to the Smoke Free app. CG has been a paid scientific consultant for the behaviour change and lifestyle organization, One Year No Beer, and provided fact checking for blog posts. CL and TO have no competing interests to declare.

promoting autonomy, and integrating interruptive and ambient content while also meeting stakeholder needs and expectations in terms of privacy.

## Author summary

Just-in-time adaptive interventions (JITAIs) are new digital tools that can provide personalised, real-time support for people trying to change their behaviour. To help the research community and developers understand how a JITAI could help people stop smoking, the authors conducted focus group discussions with adult smokers and smoking cessation professionals. They explored how a JITAI could fit into the daily lives and care of people trying to quit smoking, what it should do, and what people thought about data collection and privacy. The findings indicate that a JITAI needs to strike a balance between being easy to use and providing enough disruption to help smokers change their behaviour. People felt that a JITAI should be flexible and personalised. They wanted it to act like a supportive friend, without being controlling. They also thought it should include both immediate, on-the-spot assistance and more detailed, reflective content. Privacy was an important concern for some, particularly for professionals, although some smokers were less concerned about this issue. Overall, these findings suggest that JITAIs need to be thoughtfully designed to respect user preferences, offer flexible and varied forms of support, and address privacy concerns to support people trying to quit smoking.

## Introduction

Smoking remains a key contributor to morbidity and mortality in the UK [1,2]. New or improved interventions are needed to increase smoking cessation rates and improve population health. Digitally mediated just-in-time adaptive interventions (JITAIs) are a novel type of behavioural intervention that may increase quit rates by filling gaps in the existing treatment landscape. Most existing behavioural interventions tend to focus on increasing general (instead of momentary) motivation to quit smoking and teaching strategies to avoid and cope with smoking urges [3]. However, this appears insufficient to prevent relapse [4]. Additionally, people trying to quit smoking often fail to actively seek out support when they most need it [5,6]. Ensuing lapses frequently trigger full relapses [7–10]. JITAIs may address this problem by prompting engagement with intervention strategies at detected moments of need [11,12].

JITAIs work by collecting data on tailoring variables at frequent intervals through active (e.g., repeated surveys assessing experiences and behaviours in naturalistic setting – also known as ecological momentary assessments (EMAs)) or passive (e.g., geo-sensors, heart-rate monitors, or accelerometers) means to detect opportune moments for intervention delivery [12]. As predictors of smoking lapses during quit attempts vary over time and are often momentary [13], JITAIs that collect a wealth of data on these variables

may be able to provide tailored support at the right moment and stop lapses from occurring or escalating into full relapses. However, collecting the required amount of data may lead to concerns around privacy and data security [14].

JITAIs for smoking cessation are still in their infancy and have, so far, only been implemented using apps [15–21]. These existing app-based JITAIs do however differ in how they collect data on tailoring variables, how they determine opportune moments to intervene, the kinds of intervention they deliver at these opportune moments, and the wider context they are part of [15–21]. They seem to be broadly acceptable and feasible [15–17,19–21], and are promising in terms of effectiveness [20]. However, existing JITAIs for smoking cessation have only used some of the specific capabilities JITAIs offer, and at this point, how to most effectively implement smoking cessation JITAIs and make them acceptable and attractive to a broad range of potential users remains underexplored. So far, no smoking cessation JITAI has combined data from different sources (e.g., from both EMAs and geo-sensors) to predict and prevent lapse events or used interventions beyond text-based notifications containing coping prompts. Additionally, all but one [19,20] of these JITAIs have been embedded within larger cessation platforms or programmes. Additionally, there is little research on stakeholder needs, expectations, and preferences for smoking cessation JITAIs specifically. Since JITAIs function differently from other digital interventions and given the ever-changing landscape of digital interventions, previous work on other types of digital interventions may be of limited applicability. Therefore, there are many open questions about the intervention platforms, features and content, and integrations with other interventions that potential users prefer or want, as well as the concerns they have around issues like privacy and security.

A systematic intervention development process is crucial for complex interventions such as JITAIs to design interventions that are acceptable, feasible, and address the needs of their users and other stakeholders [22,23]. Specifically, it is important to understand the behaviour of interest (in this case, smoking cessation) in its context and how an intervention with certain capabilities (in this case, a JITAI) may fit into the lives of potential users of the intervention [23,24]. It is also important to directly address concerns potential users may have [25]. Involving users and other stakeholders in intervention design and development can allow researchers to leverage their knowledge and enables mutual learning and collective creativity [26–29]. This study involves two groups of stakeholders, namely smokers and smoking cessation professionals, due to the complementary insights they can offer on how JITAI could or should be implemented [30]. Smokers can provide insight into the lived experience of quitting and use of support, personal motivators, practices, and preferences, while smoking cessation professionals can provide insights into feasibility and adherence, common patterns, and integration with healthcare services.

## Study aims and research questions

This study aims to understand stakeholder perspectives on JITAIs for smoking cessation. Specifically, this study seeks to answer the following questions based on insights from two complementary groups of stakeholders (i) people who smoke (herein: smokers) who are motivated to quit and (ii) smoking cessation professionals who have experience working with digital services:

1. How can a smoking cessation JITAI be integrated into the lives and smoking cessation journeys of potential users?

2. What features and content do potential users value?

3. What intervention platforms do potential users prefer?

4. What kind of integration with other (digital, in-person, or pharmacological) interventions is valued and feasible?

5. What kind of data are needed for integration with other interventions?

6. What kind of data are potential users willing to provide to the intervention?

By addressing these questions, we aim to inform JITAI design by identifying preferred features (e.g., real-time support vs. ambient content) and more general design considerations and principles to meet stakeholder needs, expectations,

concerns, and preferences. The topic guides, which are attached as supporting information (S1 and S2 Files), provide more details on how the research questions informed the topic guide.

## Materials and methods

The Consolidated Criteria for Reporting Qualitative Research (COREQ) were followed in the design and reporting of the study [31]. The completed checklist can be found attached as supporting information (S3 File). The protocol for this study was pre-registered on the Open Science Framework https://osf.io/p7c9y/.

### Design

This study was designed as part of a mixed-methods project with a pragmatist epistemology [32,33]. This means that the project and this study are primarily concerned with gaining relevant, actionable knowledge from various sources that describe the problem at hand it its context but may not necessarily translate into definite scientific truths [32,33]. There-fore, this study asked questions and analysed data to directly address considerations relating to the complex, real-world intervention context and involved two groups of stakeholders with complementary perspectives. We conducted separate focus groups with two stakeholder groups: (i) current smokers motivated to quit and (ii) smoking cessation professionals with experience working with digital services. Focus groups are a suitable way to explore stakeholder perspectives on health interventions [26,30,34], particularly in the case of new and emerging interventions such as JITAIs, which partic-ipants may not have much experience with. This is because topics or ideas which may not have come up in individual interviews can emerge in the interactive, dynamic setting of a focus group [35]. Separate focus groups with separate topic guides were conducted with smokers and smoking cessation professionals. This was done to ensure that the questions were relevant and answerable for each participant [30] and to improve participants' willingness to share their viewpoints [35].

We conducted a mix of online and in-person focus groups to broaden participation. Both online and in-person focus groups may present barriers for different kinds of participants with different resources but appear to generate data of similar quality [36]. The modality for each focus group was chosen based on pragmatic considerations and participant preferences.

This study aimed to conduct three groups with smokers and one with smoking cessation professionals, with six to nine participants per group. The numbers of groups and participants were chosen because previous research indicates that given the stratification of groups and the fact that this study seeks to capture explicit and concrete rather than conceptual codes, this would allow us to achieve an acceptable degree of saturation under present resource constraints [37]. While saturation was not formally assessed, all but one minor theme were discussed across multiple groups and no one single group substantially changed the meaning of any themes. This indicates theoretical and meaning saturation, in line with commonly used definitions [35].

The UCL Research Ethics Committee has granted ethics approval (ID:26419.001). Participants provided written informed consent.

### Recruitment and setting

Most smokers were recruited via paid-for social media ads on Facebook and Instagram, but some were also recruited via unpaid posts on X (formerly Twitter), print adverts at and around the UCL campus (Camden, Greater London), at nearby community centres, and via word-of-mouth. Smoking cessation professionals were recruited via the mailing list of the UK National Centre for Smoking Cessation and Training. Potential participants were screened using a survey to confirm eligi-bility. Smokers were eligible to participate if they were at least 18 years old, lived in the UK, declared that they smoked at all nowadays, and were motivated to stop smoking at some point in the future (operationalised as a score of three or more on the Motivation to Stop Scale [38]). Smoking cessation professionals were eligible to participate if they were at least 18

years old, currently provided smoking cessation advice to people living in the UK as part of their job, and ever provided smoking cessation advice integrated with digital services. If initial information indicated that a person was eligible, they were asked to fill out a consent form and baseline survey that assessed demographic information (e.g., age, gender, ethnicity). For smokers, the survey also assessed educational attainment, occupational social grade, smoking behaviour, and smoking history. For smoking cessation professionals, the survey also contained questions about their experience providing smoking cessation support. Details of these surveys can be found in the supporting information (S4 and S5 Files). Eligible participants were contacted by email to schedule and confirm a group slot. After issues with disengaged (e.g., not turning on their video, only participating via the chat, or not participating at all) participants during the first online focus group, eligible participants for subsequent groups were also contacted via video call to check their details ahead of confirming a slot. In line with over-recruitment recommendations, we aimed to recruit 10–12 participants for each session in order to have six to nine participants per focus group [35].

The online groups (two groups of smokers; one group of smoking cessation professionals) took place via Microsoft Teams, and the in-person group (one group of smokers) took place on campus at UCL. The focus groups were conducted between January and June 2024, lasted around 60 minutes each, and were recorded using Microsoft Teams. CL facilitated the focus groups, while TO co-facilitated, took observational notes, and monitored the recording equipment.

Each focus group began with an introduction that was aided by presentation slides, which can be found attached as supporting information (S6 and S7 Files), and lasted approximately 10–15 minutes. During the introduction, facilitators provided an explanation of what a JITAI is and can be used for, an overview of the ground rules, and quickly introduced the topics that would be covered in the focus group. This was followed by an icebreaker. After the icebreaker, the focus group discussion started, and the facilitators asked the questions in the topic guide. The facilitators encouraged participants to discuss the questions among each other, only providing prompts or asking further questions to aid the flow of the discussion. This main discussion lasted around 45 minutes. After all questions in the topic guide were discussed, the facilitators asked the participants if they had final thoughts and questions, and finally thanked the participants for their time. Participants received £20 in vouchers as compensation for participation. Refreshments were offered to participants attending the in-person focus group.

## Topic guide

Topic guides addressed the integration of a JITAI into its users' lives and care provision, preferred content and features, and data and privacy. Questions and prompts included "How do you feel about the idea of the JITAI collecting, storing, and analysing data about your smoking habits and cravings?" (smokers) and "How could a JITAI be integrated into the services and support you provide?" (smoking cessation professionals). Before finalisation, an existing public and patient involvement group affiliated with the research team reviewed the topic guides. In addition, the finalised topic guides underwent informal piloting through a mock focus group. This group comprised non-researchers known to the authors who had some experience with smoking but were largely unfamiliar with the research area. Insights from this process led to minor refinements in the wording of certain questions and adjustments to the presentation slides to improve clarity. The full topic guides and presentations shown to participants can be found in the supporting information (S1, S2, S6 and S7 Files).

## Analysis

CL manually transcribed the audiovisual recordings of the focus groups with the help of automatic transcription software. TO created conversation matrices during the focus groups that were used to organise the transcript, assess consensus in the focus group, and preserve context [30,35].

The transcripts and conversation matrices were analysed using Framework Analysis [39–41]. Framework Analysis is a flexible and transparent form of thematic analysis that works well within a pragmatist epistemological outlook [39,40] and is appropriate to address strategic research questions in the field of intervention development [41,42]. Because

data is analysed by case and theme in Framework Analysis, it facilitates comparison between units of analysis [39]. This makes it particularly well-suited to this study, as we sought to understand, compare, and contrast different stakeholder perspectives on specific aspects of implementing a JITAI for smoking cessation. We used a combination of inductive and deductive coding. Deductive codes were informed by the Theoretical Domains Framework (TDF) which is an empirically informed theoretical framework which outlines 14 domains of factors that influence behaviour and behaviour change [43] and the extended version of the second iteration of the Technology Acceptance Model (TAM2), which adapts the Theory of Reasoned Action [44] and the Theory of Planned Behavior [45] to describe and explain how and why people adopt and engage with new technologies [46,47]. This mix of deductive and inductive coding is recommended when using a theoretical framework to guide qualitative analysis, particular in the field of intervention development [48]. CL led the analysis and created the initial codes and themes. The codes, themes, and their relation to one another were then discussed and agreed upon by the whole research team. A codebook describing the themes and their definitions was iteratively developed in discussion with the whole research team. This collaborative process ensured that the coding framework was coherent, clearly defined, and consistently applied, thereby enhancing the reliability of the analysis mainly conducted by one coder.

A high-level summary of results was returned to a subsample of participants for checking. NVivo was used to manage and analyse the focus group data. Quotations are provided to illustrate findings and more richly describe themes [49].

### Research team and reflexivity

CL is a female PhD student in her mid-twenties. Her project focuses on the development of a JITAI for smoking cessation. She has an MSc in Health Psychology and has previously co-facilitated focus groups and conducted thematic analysis. CL has never smoked regularly but has used cigarettes and other tobacco and nicotine products occasionally. She has never used a JITAI but has previously used other types of digital health interventions. TO is also a female PhD student in her mid-twenties. Her project focuses on the development of a virtual reality intervention for smoking cessation. She has an MSc in Public Health and has previous experience and training in conducting interviews and focus groups and in qualitative analysis. TO has never smoked or used other tobacco or nicotine products. She has never used a JITAI before but has used other types of digital health interventions. Therefore, the research team's experience with smoking and smoking cessation is primarily academic rather than based on individual lived experience, and they have a general belief that JITAIs and digital smoking cessation interventions more broadly have the potential to improve public health and are worth exploring. Although the team tried to be led by participants and the data in their analyses, this will likely have informed their view when analysing the data.

Study participants did not know the focus group facilitators or the wider research team and had little contact before the focus groups except for a reminder email or video call. The research goals were briefly reiterated to the participants at the start of each focus group session. However, care was taken not to overemphasise them to reduce potential social desirability effects.

## Results

### Participants

**Smokers.** In total, N = 431 individuals completed the screening survey to participate in the focus groups with smokers. However, N = 409 individuals were excluded (see Fig 1 for the reasons). N = 22 participants were invited to attend the three focus groups. However, N = 3 cancelled or failed to attend on the day, meaning that N = 19 participants attended across three groups (see Fig 1).

More men than women were included, and the sample was relatively young and ethnically diverse (see Table 1). Most participants had post-16 educational qualifications, but occupations were more diverse. Most participants were not parents or carers. Furthermore, most participants had a low heaviness of smoking, as indicated by the Heaviness of Smoking

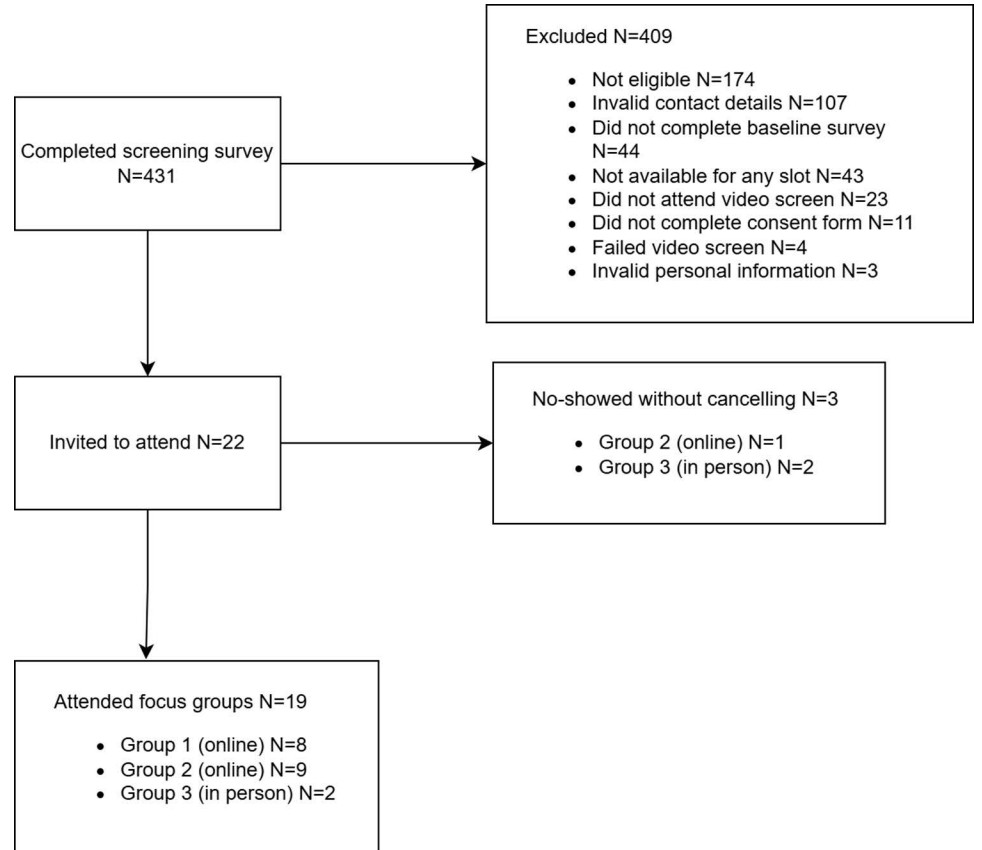

**Fig 1. Selection of participants for the focus groups with smokers.**

Index [50,51], and a low motivation to quit smoking, as indicated by a score below five on the Motivation to Stop Scale [38]. Previous quitting experience varied, with about a third having never tried to quit smoking before and about a quarter having tried to quit in the past year. The majority had not previously used digital smoking cessation support but about a third of participants had used some type of digital behaviour change intervention.

**Smoking cessation professionals.**  In total, N = 83 individuals completed the screening survey to participate in the focus group with smoking cessation professionals. However, N = 74 individuals were excluded (see Fig 2 for the reasons). Of the N = 9 smoking cessation professionals invited to attend, one cancelled via email and three cancelled or failed to attend on the day, leaving a final sample of N = 5 smoking cessation professionals (see Fig 2).

Three out of five participants were women, and the overall sample was relatively young and ethnically diverse (see Table 1). Most participants worked in London, and all were employed by the UK's National Health Service (NHS), with one working in a prison setting. Smoking cessation support formed the majority of most participants' workload.

## Overview of themes

Overall, our analysis identified four major themes, "Smoking Cessation Process", "JITAI Characteristics", "Perceived Value", and "Relationship with the JITAI", along with 16 subordinate themes (see Fig 3; S8 File for codebook). Although the theme "JITAI Characteristics" was most frequently discussed, none of the themes was dominant; rather, the themes interlinked and informed each other. Smoking cessation professionals and smokers raised largely similar themes, although some of the specific perspectives, particularly regarding user engagement and privacy and data protection

PLOS Digital Health

**Table 1. Demographic, smoking, and professional characteristics of participants across focus groups.**

| Demographic, smoking, and professional characteristics | | Smokers overall (N = 19) | Smokers group 1 (online; N = 8) | Smokers group 2 (online; N = 9) | Smokers group 3 (in person; N = 2) | Smoking cessation professionals (online; N = 5) |
|---|---|---|---|---|---|---|
| Length (minutes) | | ----- | 69 | 78 | 42 | 58 |
| Gender; n (%) | Men | 13 (68.4) | 6 (75) | 6 (66.7) | 1 (50) | 2 (40) |
| | Women | 6 (31.6) | 2 (25) | 3 (33.3) | 1 (50) | 3 (60) |
| | In another way | 0 (0) | 0 (0) | 0 (0) | 0 (0) | 0 (0) |
| Age (years) | Mean (median; range) | 29.7 (27; 20-65) | 25.1 (24.5; 20-30) | 33.8 (31; 22-65) | 30 (30; 22-38) | 30.6 (29; 29-34) |
| Ethnicity; n (%) | Any Asian or Asian British | 3 (15.8) | 0 (0) | 2 (22.2) | 1 (50) | 3 (60) |
| | Any Black, Black British, Caribbean, or African | 10 (52.6) | 7 (87.5) | 3 (33.3) | 0 (0) | 1 (20) |
| | Any white | 6 (31.6) | 1 (12.5) | 4 (44.4) | 1 (50) | 1 (20) |
| Occupational status; n (%) | Manual | 4 (21.1) | 4 (50) | 0 (0) | 0 (0) | ----- |
| | Non-manual | 9 (47.4) | 1 (12.5) | 7 (77.8) | 1 (50) | ----- |
| | Full-time student | 5 (26.3) | 3 (37.5) | 1 (11.1) | 1 (50) | ----- |
| | Unemployed | 1 (5.3) | 0 (0) | 1 (11.1) | 0 (0) | ----- |
| Educational attainment; n (%) | Post-16 qualifications | 17 (89.5) | 7 (87.5) | 88.9 (8) | 2 (100) | ----- |
| | No post-16 qualifications | 2 (10.5) | 1 (12.5) | 1 (11.1) | 0 (0) | ----- |
| Caring responsibility; n (%) | Not a carer | 15 (78.9) | 4 (50) | 9 (100) | 2 (100) | ----- |
| | Parent | 3 (15.8) | 3 (37.5) | 0 (0) | 0 (0) | ----- |
| | Carer of a disabled person | 0 (0) | 0 (0) | 0 (0) | 0 (0) | ----- |
| | Carer of an elderly person | 1 (5.3) | 1 (12.5) | 0 (0) | 0 (0) | ----- |
| Heaviness of Smoking Index (scale 0–6; low 0–1, moderate 2–4, high 5–6); n (%) | Low | 15 (78.9) | 7 (87.5) | 6 (66.7) | 0 (0) | ----- |
| | Moderate | 3 (15.8) | 0 (0) | 3 (33.3) | 2 (100) | ----- |
| | High | 1 (5.3) | 1 (12.5) | 0 (0) | 0 (0) | ----- |
| Motivation to Stop Scale (scale 3–7; higher ≥ 6); n (%) | Higher | 2 (10.5) | 2 (25) | 0 (0) | 0 (0) | ----- |
| | Lower | 17 (89.5) | 6 (75) | 9 (100) | 2 (100) | ----- |
| Previous quit attempts; n (%) | Never | 6 (31.6) | 3 (37.5) | 2 (22.2) | 1 (50) | ----- |
| | Yes, but not in the past year | 8 (42.1) | 2 (25) | 5 (55.6) | 1 (50) | ----- |
| | In the past year | 5 (26.3) | 3 (37.5) | 2 (22.2) | 0 (0) | ----- |
| Previous use of cessation aids; n (%) | Pharmacotherapy | 9 (47.4) | 3 (37.5) | 6 (66.7) | 0 (0) | ----- |
| | Vapes or heated tobacco products | 9 (47.4) | 3 (37.5) | 5 (55.6) | 1 (50) | ----- |
| | Digital support | 2 (10.5) | 0 (0) | 2 (22.2) | 0 (0) | ----- |
| | Other behavioural support | 7 (36.8) | 4 (50) | 3 (33.3) | 0 (0) | ----- |
| | Alternative medicine | 0 (0) | 0 (0) | 0 (0) | 0(0) | ----- |
| | None of these | 5 (26.3) | 3 (37.5) | 1 (11.1) | 1 (50) | ----- |
| Previous use of digital health; n (%) | Yes | 7 (36.8) | 4 (50) | 3 (33.3) | 0 (0) | ----- |
| | No | 12 (63.2) | 4 (50) | 6 (66.7) | 2 (100) | ----- |
| Percentage of workload taken up by smoking cessation support | < 50 | ----- | ----- | ----- | ----- | 1 (20) |
| | 50-75 | ----- | ----- | ----- | ----- | 1 (20) |
| | 75-100 | ----- | ----- | ----- | ----- | 3 (60) |

*(Continued)*

Table 1. (Continued)

| Demographic, smoking, and professional characteristics | | Smokers overall (N = 19) | Smokers group 1 (online; N = 8) | Smokers group 2 (online; N = 9) | Smokers group 3 (in person; N = 2) | Smoking cessation professionals (online; N = 5) |
|---|---|---|---|---|---|---|
| Region of work; n (%) | London | ----- | ----- | ----- | ----- | 4 (80) |
| | West Midlands | ----- | ----- | ----- | ----- | 1 (2) |
| Employer; n (%) | NHS | ----- | ----- | ----- | ----- | 5 (100) |
| | Prison | ----- | ----- | ----- | ----- | 1 (20) |

diverged between stakeholder groups. Although the analysis was guided by the TDF [43] and TAM2 [46,47], most themes in the framework emerged inductively during familiarisation with the data and construction of the analytical framework.

### Smoking cessation process

The major theme "Smoking Cessation Process" has four subordinate themes, "Time and Stability", "Emotions", "Beliefs about Capabilities", and "Social Influences". "Smoking Cessation Process" was coded inductively. Participants largely described smoking cessation as an idiosyncratic, nonlinear, and emotionally challenging process that the JITAI should accommodate and support people throughout. However, one participant described quitting smoking as more akin to the flipping of a switch.

*"I suppose that kind of aligns with your own quitting goals, doesn't it? So if you say I want to quit within X amount of months, you'd want to have it, for, for that amount of time, so I suppose it would vary from person to person." (Smokers Group 3, 38M, non-manual occupation)*

*"[U]ltimately you can't say 'Well, I've used it for three months, I've stopped smoking, I'm okay' and then six months down the line, you pick up a cigarette. It might possibly be that it can not only help you to stop, but cope with, umm, everything that comes along for maybe even years." (Smokers Group 2, 65M, non-manual occupation)*

**Time and stability.** "Time and Stability" was coded inductively. Overall, participants noted that one's motivation to quit smoking and remain abstinent was changeable and subject to various influences and that the timescale of one's quit attempt could vary substantially between individuals. There was consensus a JITAI should accommodate these individual differences.

*"I was assuming that minimum three months maximum a year because everybody has a different healing process, some might be faster. Some might be a bit slow, but making this service available for everyone, everyone should use depending on their speed and how willing they are, able to do this and how consistent they can be. That would be really nice." (Smokers Group 1, 23 F, routine or manual occupation)*

*"I think it would depend on how long I had the cravings for and how useful the app is. If it's enjoyable to use and I've found it helpful then it would last longer" (Smokers Group 2, 27M, non-manual occupation)*

**Emotions and beliefs about capabilities.** The themes "Emotions" and "Beliefs about Capabilities" were deductively coded from the TDF. While they were not discussed in all groups, when they came up, participants described a bidirectional relationship between (negative) emotions and smoking cessation. They felt that a JITAI should manage and support users through these quitting-related emotions and difficulties.

*"[C]ause it's the minute that you get a little bit of adversity or a little bit of stress or something goes wrong, then that's the first thing you, you, you use as a coping mechanism. But it's the first thing the brain goes to. So, anything that is able to support and just help you with your routine and happiness" (Smokers Group 2, 34M, non-manual occupation)*

 

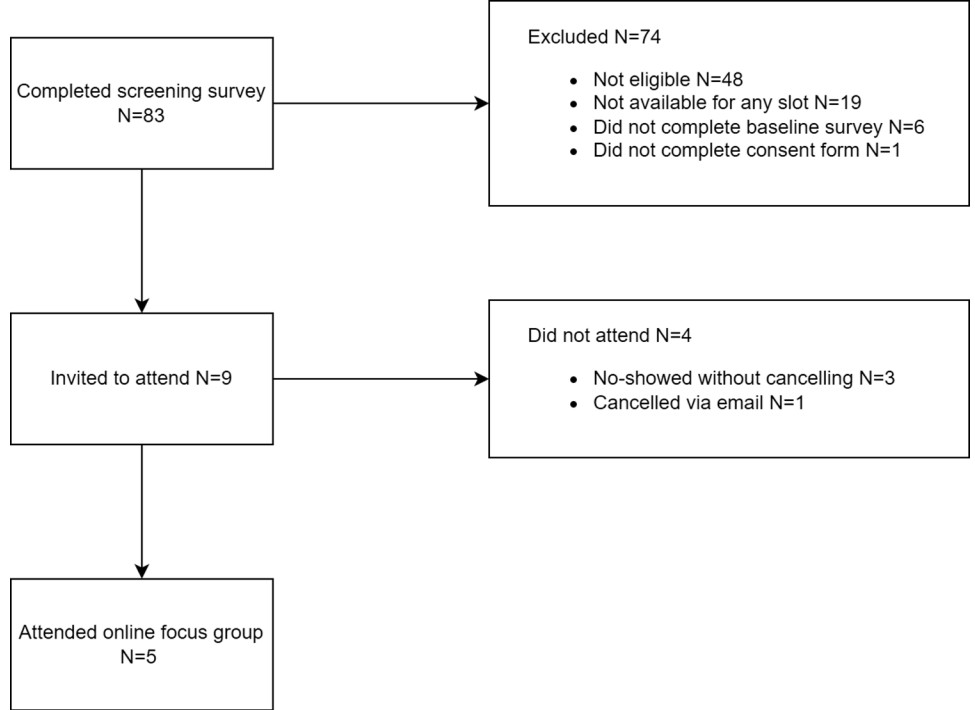

**Fig 2. Selection of participants for the focus groups with smoking cessation professionals.**

Additionally, some participants described building confidence in one's ability to quit smoking and make good decisions for oneself as an important part of the smoking cessation process.

> "[F]or me, it's really important that I get to make certain decisions first. Uh, something with smoking as a decision has limited my choices [...]. So being able to choose certain things just makes me feel like, yeah, I'm making the right decision and I'm capable of doing this by myself." (Smokers Group 1, 23F, routine or manual occupation)

**Social influences.** "Social Influences" was coded deductively from the TDF and was discussed extensively in all groups. Participants discussed both negative social influences to keep smoking and positive social support to help sustain one's quit attempt. Participants expressed interest in leveraging the latter within a JITAI, either by involving existing friends and family or creating a digital community.

> "[T]here are things that will want to drag us back, moments in our lives, places we visit, friends we stay with that will want to make us go back to these ways." (Smokers Group 1, 23M, full-time student)

> "I think what helped me during a time I tried quitting earlier and stayed away from it for a while, was that was a friend along with me, who was also trying to do the same, and the two of us would kind of look out for each other in that sense. [...] I don't know about, you know, privacy concerns, obviously, but then, could this be a set of people who have signed up for the service. Could they be those friends for each other?" (Smokers Group 2, 43M, non-manual occupation)

However, there was also some ambivalence about social support. Smoking cessation professionals highlighted that some of their patients preferred to go about their quit attempt on their own. Participants across groups also highlighted that

PLOS Digital Health

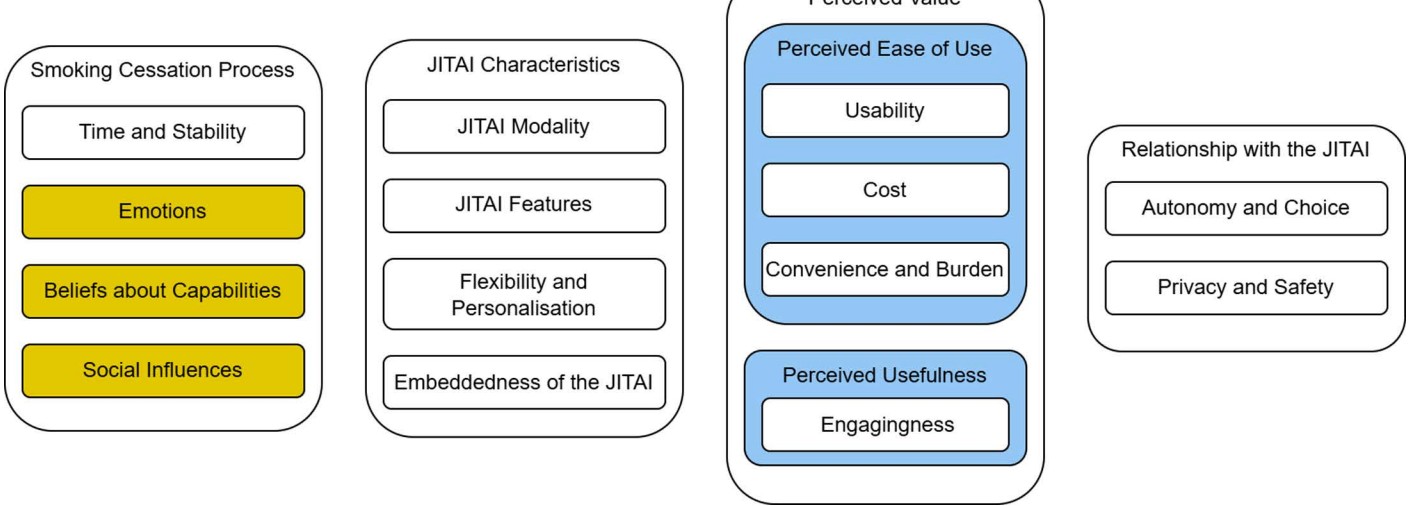

**Fig 3. Overview of hierarchically structured themes.** No fill indicates that themes were coded inductively, yellow fill indicates that themes were coded deductively from the TDF, and blue fill indicates that themes were coded deductively from the TAM2.

social support could become negative in the form of group toxicity or guilt-tripping and expressed concerns about potential safety issues arising from a digital community.

> *"[O]bviously it can work both ways. The patient doesn't really want to quit smoking, but the family member might guilt trip them to be like if you don't, you're going to die and all this like negative stuff. So obviously it works both ways." (Smoking Cessation Professionals, 32M)*

> *"I think sometimes it gets in your head and we… we make it toxic for one another." (Smokers Group 1, 23F, routine or manual occupation)*

### JITAI characteristics

The major theme "JITAI Characteristics" is an umbrella theme that comprises four subordinate themes, "JITAI Modality", "JITAI Features", "Flexibility and Personalisation", and "Embeddedness of the JITAI". It was coded inductively to characterise the four subordinate themes which were raised across all groups. Participants talked about both the concrete modalities and features that they preferred and the more abstract characteristics that influenced how and why they preferred certain modalities and features.

**JITAI modality.** "JITAI Modality" was coded inductively and was discussed in similar ways across all groups. Generally, apps were the preferred intervention modality. While participants valued apps' versatility and capacity to host multimedia content, some expressed concern that they could not be sufficiently personalised. Notifications and text messaging were considered convenient and easy to integrate into daily life, but some participants stated that they could become annoying, boring, or trigger cravings. Smartwatches were described as the most convenient intervention modality. There was also some ambivalence about using social media; while the social support they could offer was seen as beneficial, there were concerns about privacy and safety.

> *"So, I think an app would be a really good idea, especially targeting the younger population. And I think, you know, apps are easily downloadable. Yes, there might be a small proportion of people who won't be able to access it. Umm,*

*however, you know, in my experience, a lot of older people also are quite technologically savvy at the moment. […] But I think a text messaging service as an option for people who are less technologically savvy I think would be OK as well. Umm, but definitely there would a lot more to do with, with an app there." (Smoking Cessation Professionals, 34F)*

**JITAI features.**  "JITAI Features" was coded inductively. Similar topics and ideas within this theme were raised across all groups. Participants expressed a strong interest in a variety of features, with many emphasising the importance of catering to individual and situational differences. There was interest in features like notifications, multimedia content (video, audio, infographics), interactive elements, self-reflection and self-monitoring tools (journals, streak trackers), reinforcement (specifically, rewards) for abstinence, self-compassion or coping skills lessons, provision or suggestion of distraction or substitution, and proactive information about the health, financial, or environmental consequences of smoking. A recurring idea was to have proactive, interruptive support for immediate needs, while also providing ambient access to more in-depth content.

*"I think notifications initially, but I was also thinking earlier like a journal feature would be quite nice, like not as like the main part, but as just like a little sidebar of it. I think just if you got frustrated at some point and it said "Do you wanna write your feelings down?" […] but the main part being like a streak or like the notification. Or like you could just, like, do something quickly. But yeah, just as, like, a deeper bit, if you want that, that could be a good feature." (Smokers Group 2, 32F, non-manual occupation)*

**Flexibility and personalisation.**  The theme "Flexibility and Personalisation" was coded inductively. There was consensus that personalising and adapting a JITAI to individual needs, values, quitting goals, and progress was important, with many participants considering it essential. For example, participants who were parents mentioned integrating and acknowledging that their child was an important motivator for them to quit smoking would be helpful for them, but inapplicable for people who do not have children. Mood, location, and smoking history were described as potential tailoring variables.

*"Everyone's needs are different, everyone's got different jobs, different sort of habit, different stresses. So, it really should be kind of personalised and tailored to your life." (Smokers Group 2, 34M, non-manual occupation)*

*"[B]ut that could also incorporate uh your feedback. Uh, how I am feeling, what is happening with me, and direct me to the local pharmacy depending on where I am and not just not just to say to me, go to your pharmacy but "Ohh go into Boots, it's only around the corner, one minute walk and they will give you some nicotine patches or something like that, that can help you through this particular time that the phase you are going through right now because it's early days" and it can adjust of you initial, uh, requirements depending on the information that it's collecting about you." (Smokers Group 2, 65M, non-manual occupation)*

*"I suppose there'd be like a number of approaches in the app. There would be notifications and data, and this, that, and various things. So, which some people might have no interest or in one of them, or, or another. Perhaps at the beginning of your quitting journey you sort of predefine what methods you are going to be using and that might not be all of the methods within the app, so turn them off then." (Smokers Group 3, 22F, full-time student)*

**Embeddedness of the JITAI.**  The theme "Embeddedness of a JITAI" was coded inductively. Opinions on integrating a JITAI with the wider healthcare system and other interventions varied. Some participants expressed interest in integrating a JITAI with counselling, general practitioner assistance, nicotine replacement therapy, and their health records to make it more effective and increase confidence in it by connecting it to a trusted institution.

*"Using the intervention along side [sic] with a good therapist would be much efficient [sic]." (Smokers Group 1, 28F, full-time student)*

However, some also expressed reluctance about sharing their data and doubts about the usefulness of such an integration. Due to these privacy concerns, participants generally agreed that integration with the wider health system should be voluntary and flexible.

*"[I]t could be linked to the, the NHS app, but I don't want GPs [general practitioners] getting too much information about everything that you do all day long uh, just, just the important bits." (Smokers Group 2, 65M, non-manual occupation)*

*"Well, I guess once it's up and running. I mean, I'd be fine with obviously using anonymous data. [...] Some people might not be. So, they'd probably want the option to not have their data shared at all. Just, just so the, the app just works for them, no sharing of data basically to anyone" (Smokers Group 3, 38M, non-manual occupation)*

Additionally, some participants suggested integrating a JITAI into a multidisciplinary or lifestyle intervention to increase longevity and use, while others, primarily smoking cessation professionals, expressed concerns about overloading users and keeping the intervention focused. However, smoking cessation professionals also mentioned that some sort of integration with carers or primary care providers may be beneficial as an accessibility adaptation for users with disabilities or older users.

*"So, it's like a, a lifestyle app, if you were to call it that, uh, but everything is strung along together and that that's what makes it interesting […] [I]t's a one-stop app for everything that then interests me" (Smokers Group 2, 43M, non-manual occupation)*

*"Umm, I think there should be, might be, like a link to, like, extra resources or other services so they have the option of going on to that if they want, but not just like popping up like. Do you need this service, or you know if you want an appointment for that service? I don't. I don't think that'll be productive for, for them." (Smoking Cessation Professionals, 29F)*

### Perceived value

The major theme "Perceived Value" is an inductively coded umbrella theme comprising the two determinants of intention to use in the TAM2, "Perceived Ease of Use" and "Perceived Usefulness". Both have inductively coded subordinate themes of their own. Participants expressed that they would only continue using a JITAI if they perceived it to bring value to their lives. Participants' discussions indicated that the value of a JITAI would be influenced by both how easy it is to use and how useful it is.

**Perceived ease of use.** The umbrella theme "Perceived Ease of Use" was deductively coded from the TAM2 and summarises its three subordinate themes "Usability", "Cost", and "Convenience and Burden".
*Usability:* "Usability" was coded inductively. Participants expressed that a JITAI's user interface should be intuitive and not overwhelm users with content. They also stated that some privacy protections, like passwords, could impede usability.

*"I would like for it to be flexible and the dashboard should be – if it's an app I would like the dashboards to be very simple, meaning I do not have to look and they're not hidden pages where you have to move, move, move just to get the page." (Smokers Group 1, 23F, routine or manual occupation)*

*Cost:* "Cost" was coded inductively and only mentioned in one of the groups. Overall, there was a consensus that a JITAI should be free at the point of access to encourage use, ensure equity, and increase trust. However, a few participants also argued that users might be more committed to a JITAI and their smoking cessation attempt if they had to pay for it.

*"For me, I think it's about accessibility because a free service, I mean obviously like you know, buying cigarettes and et cetera. But it's just like, anybody could then access the app and then yeah. It's about accessibility at the end of the day." (Smokers Group 2, 32F, non-manual occupation)*

*"Yeah, I, I think I, I wanna agree with that and the, uh, part about it being paid for by someone else. I do disagree on that one, because I, I've seen that if something is free, then a person typically doesn't tend to kind of use it. It kind of just remains there at the back seat somewhere."* (Smokers Group 2, 43M, non-manual occupation)

***Convenience and Burden:*** "Convenience and Burden" was coded inductively. Consistently, discussions highlighted a tension between two somewhat opposing requirements participants felt a JITAI should meet: being convenient and being disruptive. Participants emphasised that a JITAI should seamlessly integrate into users' lives, provide consistently available, tailored support, and not annoy users. Both smokers and smoking cessation professionals expressed concerns about the potential burden of completing EMAs. They agreed that filling out more than a few per day would be unrealistic, with smoking cessation professionals being particularly sceptical about users completing more than two. Participants also stressed the importance of making EMAs enjoyable, short, and clearly useful. However, they also highlighted that a JITAI should be sufficiently disruptive to foster ownership of the quit attempt, engage users, and potentially enhance self-reflection.

*"I agree what he said, ultimately you've got to do something, erm, but it gives you an excuse to give up sooner if it's more difficult. [...] So, I think there's a fine balance to be found, which I suppose you've got to experiment with and find like, what's too much, what's- because you gotta do something, you gotta have a bit of involvement in the, in the thing, other it's you know, AI telling you you're doing great. But if it's something that's just a bit, you're, you're involved, but not to the point where it comes cumbersome, that that's the fine line."* (Smokers Group 2, 34M, non-manual occupation)

Overall, it appeared that the optimal trade-off between convenience and burden in a JITAI would depend on a range of parameters, including the individual, the setting, and the JITAI characteristics.

*"So, these were text messages and it just got ridiculous how often they were sending. And I wasn't reading them. It was just like, I give up. Like, I don't. I wasn't reading them, basically. So that was just too much. But yeah, once or twice a day or something that's more interactive."* (Smoking Cessation Professionals, 34F [discussing a digital health application they had used in the past for a separate reason])

**Perceived usefulness.** "Perceived Usefulness" was coded deductively from the TAM2 and was described as a key determinant of JITAI use. Participants consistently emphasised that they would only continue to use a JITAI if they could align it to their smoking cessation goals.

*"So perhaps you know, perhaps it needs a sense of being realistic, and it's only gonna work for people that actually want it to work. You, you know if, if you're gonna take it on and see how it's gonna work. And yes, if it's helping you, you're gonna carry on."* (Smokers Group 2, 65M, non-manual occupation)

***Engagingness:*** "Engagingness" was coded inductively. Although there was a broad consensus that a JITAI's engagingness (i.e., its capacity to promote engagement with it) was important, it appeared to be more critical for some participants than others. In general, participants stated that repetitiveness would diminish a JITAI's engagingness, while gamification, personalisation, opportunities for self-reflection, and multimedia interactivity would enhance it.

*"I think it needs to be initially very short term, engaging, it needs to be, like, snappy and, like, it needs, like, you need to engage with it very quickly and it needs to capture your attention very, very, well very quickly."* (Smokers Group 2, 32F, non-manual occupation)

*"I will give it a try and then maybe I'll keep it. I'll keep it there if it's not too annoying and I'm not sure, but like it's looking interesting to me." (Smokers Group 3, 22F, full-time student)*

*"So, like I said, the, the main feature for me would be I want it to be fun, I want it to be exciting." (Smokers Group 2, 65M, non-manual occupation)*

**Relationship with the JITAI**

The major theme "Relationship with the JITAI" was coded inductively. It has two subordinate themes, "Autonomy and Choice" and "Privacy and Safety". Participants consistently emphasised that they would want to feel comfortable with a JITAI, likening the preferred relationship to that with a friend or companion – supportive but not nagging, judgmental, patronising, or overly formal.

*"You won't forget that it's there ever, because that's the thing. It becomes your friend. It becomes your, whatever you want it to be, because it is there." (Smokers Group 2, 65M, non-manual occupation)*

*"But yeah, it's just, I guess, things have to be presented in a way that they're not being judged for whatever information they input into the app or service." (Smoking Cessation Professionals, 34F)*

*"It's giving you that sense of partnership, but at the same time not being overly invasive." (Smokers Group 1, 43M, non-manual occupation)*

**Autonomy and choice.** "Autonomy and Choice" was coded inductively, although it is closely related to the concept of "Voluntariness" from the TAM2. Participants highly valued autonomy within a JITAI, both for its intrinsic value and to make the tool more effective. There was a strong consensus that a JITAI should support, not control, the smoking cessation attempt.

*"So, it's a question of the involvement, has to be voluntary from me if I want to do it. And if I don't want to do it, whether I'm doing well or not doing well, it's up to me. I should be in control of everything." (Smokers Group 2, 65M, non-manual occupation)*

*"[F]or me, it's really important that I get to make certain decisions first." (Smokers Group 1, 23F, routine or manual occupation)*

Furthermore, participants agreed on the importance of clear consent procedures, choice, and transparency about data collection and its purpose. However, participants also raised concerns about "alibi consent forms" that were more of a tick-box exercise and might obscure rather than clarify terms and conditions.

*"Umm, maybe for every question asked there should be a prefer not to say button so they don't have, they're not obliged to answer." (Smoking Cessation Professionals, 29F)*

*"Not, not produce a great big, A4 size page full of gobbledygook that actually somewhere includes the fact that whether you like it or not, we put cookies on your device and whether you use it or not, we're gonna use it" (Smokers Group, 65M, non-manual occupation)*

**Privacy and safety.** "Privacy and Safety" was coded inductively. There was some dissent in this theme. Smoking cessation professionals were unanimously very concerned about privacy and safety, and while most smokers also highly valued privacy, they expressed more ambivalence about its importance. In terms of data collection, a tension between

minimising user burden and maintaining privacy while ensuring intervention effectiveness emerged. Overall, comfort levels with data sharing varied. Some participants were particularly reluctant to allow passive data collection or share specific types of data, such as location, potentially identifying information, ethnicity, or gender, because they felt that their safety might be compromised if these data were leaked or available to others or because they felt that it was not relevant. Additionally, participants stated that they would be more willing to share data if a JITAI was associated with a trusted institution, was transparent, and provided a good explanation for why it was asking for certain data.

> "And so I would probably hinge it to the NHS if that was possible. If it seems like it's a private business, then you'd worry that your data will be sold on and, and used in other ways, advertisement, and it's gonna be used for profit. If it's government, you then gonna get people that aren't gonna do it 'cause they don't trust the government, don't want to know them where they're going, or whatever they feel, it's invasive." (Smokers Group 2, 34M, non-manual occupation)

> "You could specify why you need that data, so if you need ethnicity, you just want to know how you better to support [sic] that population group […] So, then they might be more agreeable to share their information." (Smoking Cessation Professionals, 29F)

> "For real I would feel more comfortable if I knew exactly what data is being collected and how it's being used. I think there should be clear guidelines around data privacy and security, and people should have the option to delete their data if they wish." (Smokers Group 1, 20M, full-time student)

However, some smokers also expressed a level of privacy cynicism, arguing that no information was truly private anyway and that they were, therefore, not concerned about sharing it, particularly if they got something out of the intervention in return.

> "Personally, I don't care. I don't care who has my data, what they use it for, how they use it, because I am aware that many, many companies, government departments and everybody out there does it anyway." (Smokers Group 2, 65M, non-manual occupation)

## Discussion

### Summary of findings

Our focus groups with smokers and smoking cessation professionals indicate that to support and manage its users through the idiosyncratic and challenging smoking cessation process, a JITAI needs to balance being convenient enough to easily integrate into its users' daily lives with being disruptive enough to facilitate behaviour change. To do this, it needs to be flexible, personalisable, and consistently supportive in a way that preserves users' autonomy. Smokers and smoking cessation professionals were open to a variety of features, content, and intervention platforms. They emphasised that these characteristics should be chosen to provide an engaging experience and allow for the provision of both immediate, interruptive support and ambient, in-depth content. Participants had mixed opinions about integrating a JITAI with other interventions or the wider healthcare system and emphasised that any integration should be voluntary. Autonomy was highly valued. Privacy and safety were key concerns for smoking cessation professionals, while smokers' opinions were more mixed, with some displaying a level of privacy cynicism.

### Links and comparison to literature

The findings of this study suggest that user preferences for smoking cessation JITAIs largely align with preferences and recommendations for digital smoking cessation interventions more generally. The emphasis on the importance of personalisation, engagement strategies, and user autonomy is in line with previous research [52–59]. However, we also identify

novel considerations, such as the inherent tension of balancing convenience with sufficient disruption to support behaviour change, which has not been discussed in prior research. Additionally, while most prior research consistently highlights privacy as a major concern [54,60], our findings suggest more varied user perspectives, with some smokers exhibiting privacy cynicism and a willingness to trade data for perceived benefits.

In line with prior recommendations for the design of digital smoking cessation interventions, our findings indicate support for the use of gamification, interactivity, multimedia content, continuous novelty, and integrated gratification and rewards to enhance engagement and enjoyment of intervention use [52–54]. These findings also align with recommendations from design research that highlight the importance of creating the perception of a relationship, ambient information, and supporting emotional wellbeing along with behaviour change [55]. Participants' desire to have a JITAI act as a friend or companion also mirrors previous qualitative work [52]. The findings from this study also support previous recommendations that digital smoking cessation interventions should be designed according to principles of transparency and autonomy [53]. Previous research highlights that transparency and agency might be particularly important for underserved and marginalised groups [25], meaning that incorporating these principles may be an avenue to promote health equity. Furthermore, our analyses suggest that potential users highly value personalisation and the ability to flexibly fit an intervention into one's life, which JITAIs can enable to a greater extent than previous digital smoking cessation interventions. This aligns with the emphasis put on personalisation in previous literature and recommendations [52–54]. In terms of personalisation, previous research indicates that users appreciate interventions that are tailored to their values or culture [56] and that, based on their personality, people are differentially responsive to different persuasion strategies [57]. Tentative evidence also suggests that tailoring and targeting may make interventions more effective [61,62]. Enabling autonomy and personalisation may also help address health inequalities by allowing users to tailor a JITAI to their specific preferences and circumstances. The ability to customise the content or modality or to restrict data-sharing can reduce barriers to engagement, particularly for underserved populations who may face accessibility challenges or distrust in health services [25,58,59]. Participants in this study expressed ambivalence about integrating social connectivity into a JITAI, highlighting both the potential benefits and the potential pitfalls. A previous review has reported similar ambivalence around social integrations [54]. Although recommendations have nevertheless tended on the side of including such features [52,54], previous research indicates that existing digital smoking cessation interventions do not make full use of social engagement features [63]. Participants in this study suggested and endorsed multiple ways of integrating social connectivity into a JITAI, however, they also stated that these features should be available but not mandatory.

Participants in this study felt that a JITAI should offer a wide range of features, acknowledging that not all of these would be relevant to every user. This preference aligns with prior qualitative research, where users described health behaviour apps as a "toolbox" from which they could select what suited their needs [64]. This approach is also supported by quantitative research, which found that commercial weight management apps with more features and, to a lesser extent, more behaviour change techniques had higher user ratings [65]. Furthermore, our findings suggest that the specific features users would value in a smoking cessation JITAI are largely those that are recommended for digital smoking cessation interventions more generally, such as goal-setting, self-tracking, reminders, progress visualisations, information about consequences of (not) smoking, and craving management and coping support [52,54]. Concerns, like the worry that support messages might accidentally trigger cravings, especially among light smokers, also mirror concerns raised in previous qualitative research [66]. Overall, participants in this study did not seem to view JITAIs as new types of interventions per se, but as enhancements to make existing interventions work better.

Previous research found that participants raised concerns about the burden that self-tracking may entail [54] and has emphasised keeping interactions short and demands on users at a minimum [53]. However, findings from this study paint a more nuanced picture that suggests that although it is important to keep user burden low and enable short and timely support and interaction, some level of burden may be helpful in facilitating and supporting a quit attempt. We did not find any previous research that highlighted this same tension between being convenient enough to easily integrate

into its users' daily lives, yet disruptive enough to facilitate behaviour change. While it is unclear at which point an optimal trade-off would be reached, both participant comments and previous empirical research on the acceptability of smoking cessation JITAIs suggest that the perceived optimum may depend on the output of the JITAI [17]. Balancing these competing demands in the design and implementation of JITAIs may be difficult. One potential solution is to implement user-controlled engagement settings, allowing for the customisation of the maximum or minimum notification frequency or the content delivery modes. Additionally, adaptive algorithms that adjust intervention delivery based on user response patterns could help maintain engagement. Finally, opt-in reinforcement strategies, such as gamification and milestone-based rewards, could be used as tools to influence the convenience-burden trade-off calculus.

Most existing smoking cessation JITAIs have been integrated with other intervention modalities to some extent [15–18,21], although one recent one has been self-contained [19,20]. While some qualitative research indicates that digital smoking cessation interventions are often used as an adjunct to other smoking cessation interventions [52], in observational population studies, very few participants report using more than one quit aid at any one time [67,68]. There is also no clear indication that doing so increases quit rates [68]. There is, however, some tentative evidence that more embeddedness with counsellors and peer support may be linked to higher use of digital interventions [69]. Reviews on combined face-to-face and digital interventions in the fields of mental health and (health) behaviour change indicate that although these interventions may be promising, there is so far no clear indication that they are more effective than either solely face-to-face or solely digital interventions [70–73]. Research on mental health practitioner perspectives on integrating EMAs into clinical practice indicates that although they generally regard EMAs as useful, they are also concerned about the effort, resources, and potential adverse effects on patients from the EMA regime itself [74]. Additionally, they expressed some uncertainty about how to effectively integrate EMAs into conventional face-to-face mental health treatment [74], mirroring some of the scepticism of participants in this study. Our participants' suggestion that linking a JITAI to a reputable organisation or clinical endorsement, instead of a commercial company, might increase trust is in concordance with previous research [60,75,76]. However, previous research has shown that the majority of smoking cessation apps that are available to consumers through conventional means are created by and affiliated with commercial companies [63] and only adhere to clinical guidelines to a limited extent [77].

While previous research and recommendations have mostly emphasised the importance of privacy in digital health interventions [54,60], our participants seemed to feel more ambivalent about this topic, with some displaying a degree of privacy cynicism [78]. Generally, smoking cessation professionals seemed to place a higher importance on privacy than many potential users themselves. Some previous research on opportunities and challenges for smartphone apps for health behaviour change, which was conducted over a decade ago, has described similar user perspectives [76]. Compared to that study, similar concerns were raised in this study, such as the dislike of using health data to target advertising and personal safety issues that may arise from the sharing of location data [76]. However, there was also some divergence, with participants in this study being more enthusiastic (or at least ambivalent) about context sensing and social media integrations. This may reflect changes in the technology that is available and that participants are used to over this time. Additionally, our participants' explicitly stated willingness to trade off privacy against a service or other benefits, as a good to be exchanged rather than as a right mirrors what has been described as the commodification of privacy with the advent of the internet [79]. Participants' privacy cynicism reflects what some problematise as this dynamic occurring within unequal power relations, but without much popular resistance [79].

By conducting focus groups with two different stakeholder groups, we were able to compare their perspectives. While the groups largely agreed on most topics, there was notable divergence on some topics, specifically privacy and safety, which seemed to be of greater concern to smoking cessation professionals than to smokers. Additionally, smoking cessation professionals seemed to anticipate more issues around implementation, raising concerns around usability and expressing pessimism about the amount of disruption users would tolerate, while smokers seemed less concerned about these issues. Previous research into the implementation of other types of smoking cessation interventions indicates that

service providers may at times anticipate and emphasise barriers that do not materialise for a majority of users [80]. This indicates that while it is likely important to consider these concerns, they do not need to be regarded as inherently prohibitive to an intervention's implementation.

### Limitations and directions for future research

Although our focus groups provided valuable insights, this study also has several limitations. During our online focus groups with smokers, some participants appeared disengaged and did not turn on their cameras. However, this issue was mitigated (though not eliminated) by introducing a video call pre-screening with participants after the first group. Additionally, our sample was UK-based and skewed towards a young and highly educated demographic, although it was ethnically diverse. As a result, our sample may not have fully captured the barriers to JITAI use that older adults or individuals with lower educational attainment might face. This skew may limit generalisability as there is an existing digital divide, i.e., socioeconomic differences in access and barriers to the use of digital (health) interventions [81]. Previous research indicates that digital health interventions are disproportionately used by people in positions of social advantage [67,82], although some evidence also suggests that this relationship may be more complex than a straightforward linear association [83]. Therefore, even digital health interventions that are overall beneficial to population health may widen health inequalities [84], particularly when developed without meaningful stakeholder involvement. However, some previous research also suggests that digital smoking cessation interventions have the potential to be more effective for individuals in disadvantaged socioeconomic positions than for those in more advantaged groups [67,85]. Our sample was also limited to people with reliable internet access or the ability to commute to central London, who spoke English fluently, and who did not require any assistive technology to participate. While smoking cessation professionals raised issues related to access for populations who would not be able to participate in a focus group scenario such as this, not hearing from people directly affected means that crucial concerns relevant to them might have been missed. A further limitation is that participants generally reported lower levels of smoking and lower levels of motivation to quit. While this partly reflects the changing characteristics of smokers in England [86,87], it also means our findings may not generalise to people who smoke at higher levels or who have a higher motivation to quit. Another potential limitation is combining data from smoking cessation professionals and smokers, who have different motivations and experiences. However, similar and complementary themes emerged in analysing the data from these two separate stakeholder groups, allowing a more well-rounded picture of stakeholder and user needs, expectations, and preferences for JITAI design and implementation to emerge. Additionally, multiple participants mentioned attaching a potential JITAI to the NHS, an institution that has a unique role within British society and cultural, political, and social discourse [88,89]. Therefore, it is reasonable to assume that certain preferences, especially those concerning the affiliation and embedding of a JITAI, will not be generalisable beyond the British context. Another important consideration is the rapid evolution of technologies such as JITAIs. As these technologies develop, improve, and become more widespread, it is likely that stakeholders' opinions on them will change [90]. Therefore, the longevity of our findings is uncertain. Furthermore, the fact that qualitative research is an inherently interpretive exercise means that, as mentioned above, although the team tried to be led by participants and the data in their analyses, this will likely have informed their view when analysing the data.

Future research should aim to address some of this study's limitations by engaging more diverse samples in terms of age, socioeconomic status, and geography. Additionally, supplementing focus groups with user behaviour studies will be important to capture real-world interactions with technology. This will provide a more comprehensive understanding of how users engage with JITAIs and how interventions can be further refined to optimise both usability and effectiveness.

### Implications for practice

The findings of this study suggest that JITAI developers could prioritise personalisation and flexibility to create engaging, companion-like interventions. These elements, along with transparent data policies, could also support stakeholder

priorities such as autonomy and privacy. Additionally, developers should carefully balance convenience with sufficient disruption to drive behaviour change, tailoring this balance to the specific intervention and target population. While social support features and healthcare integration may be of value, the findings of this study suggest these should remain optional rather than essential components.

In the long term, JITAIs have the potential to make smoking cessation support more dynamic, responsive, and adaptive to individual user needs. Future advancements in machine learning and passive sensing could further refine JITAIs' ability to predict relapse risk and deliver tailored interventions in real time. However, ethical considerations regarding data use, user autonomy, and accessibility will need to be continually reassessed to ensure these innovations do not exacerbate health inequalities. However, it is important to note that JITAIs need not be and, by participants in this study, were not described as replacements for conventional face-to-face smoking cessation services but rather as additions that may make digital interventions more effective for those who choose to use them. As part of a comprehensive set of support options, JITAIs have the potential to complement existing services and provide additional, flexible assistance to those who may not otherwise engage with traditional stop-smoking programmes.

## Conclusion

This study is one of the first studies to qualitatively explore different stakeholders' perspectives on JITAIs for smoking cessation. While some of the findings align with existing research on other types of digital interventions, such as the importance of personalisation, features to promote engagement, and user autonomy, the study also highlights some previously unexplored considerations, such as a tension between convenience and sufficient disruption to support behaviour change. Additionally, the findings indicate that stakeholder perspectives on privacy are complicated: while privacy is highly valued by smoking cessation professionals and some smokers, some smokers also display privacy cynicism and a willingness to trade privacy for perceived effectiveness or ease of use.

## Supporting information

**S1 File. Topic guide smokers.**
(DOCX)

**S2 File. Topic guide smoking cessation professionals.**
(DOCX)

**S3 File. COREQ checklist.**
(PDF)

**S4 File. Screening and baseline survey smokers.**
(DOCX)

**S5 File. Screening and baseline survey smoking cessation professionals.**
(DOCX)

**S6 File. Slides smokers group.**
(PPTX)

**S7 File. Slides smoking cessation professionals group.**
(PPTX)

**S8 File. Codebook.**
(DOCX)

## Acknowledgments

Thank you to the Patient and Public Involvement (PPI) group and all participants for their invaluable contributions to this research. We also want to thank Andy McEwen for allowing us to recruit participants through the National Centre for Smoking Cessation and Training (NCSCT) mailing list.

## Author contributions

**Conceptualization:** Corinna Leppin, Claire Garnett, Olga Perski, Jamie Brown.

**Data curation:** Corinna Leppin, Tosan Okpako.

**Formal analysis:** Corinna Leppin.

**Funding acquisition:** Jamie Brown.

**Investigation:** Corinna Leppin, Tosan Okpako.

**Methodology:** Corinna Leppin.

**Project administration:** Claire Garnett, Olga Perski, Jamie Brown.

**Supervision:** Claire Garnett, Olga Perski, Jamie Brown.

**Writing – original draft:** Corinna Leppin.

**Writing – review & editing:** Corinna Leppin, Tosan Okpako, Claire Garnett, Olga Perski, Jamie Brown.

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
