## [Decision Letter · Decision Letter 0]

12 Feb 2025

PDIG-D-24-00522Exploring perspectives on digital smoking cessation just-in-time adaptive interventions: A focus group study with adult smokers and smoking cessation professionalsPLOS Digital Health Dear Dr. Leppin, Thank you for submitting your manuscript to PLOS Digital Health. After careful consideration, we feel that it has merit but does not fully meet PLOS Digital Health's publication criteria as it currently stands. Therefore, we invite you to submit a revised version of the manuscript that addresses the points raised during the review process. Please submit your revised manuscript within 60 days Apr 13 2025 11:59PM. If you will need more time than this to complete your revisions, please reply to this message or contact the journal office at digitalhealth@plos.org. Please include the following items when submitting your revised manuscript:* A rebuttal letter that responds to each point raised by the editor and reviewer(s). You should upload this letter as a separate file labeled 'Response to Reviewers '. This file does not need to include responses to any formatting updates and technical items listed in the 'Journal Requirements' section below.* A marked-up copy of your manuscript that highlights changes made to the original version. You should upload this as a separate file labeled 'Revised Manuscript with Track Changes '.* An unmarked version of your revised paper without tracked changes. You should upload this as a separate file labeled 'Manuscript '. If you would like to make changes to your financial disclosure, competing interests statement, or data availability statement, please make these updates within the submission form at the time of resubmission. Guidelines for resubmitting your figure files are available below the reviewer comments at the end of this letter. We look forward to receiving your revised manuscript. Kind regards, Haleh AyatollahiSection EditorPLOS Digital Health Haleh AyatollahiSection EditorPLOS Digital Health Leo Anthony CeliEditor-in-ChiefPLOS Digital Healthorcid.org/0000-0001-6712-6626 **Additional Editor Comments (if provided):****Reviewers' Comments:** Reviewer's Responses to Questions

**Comments to the Author**

1. Does this manuscript meet PLOS Digital Health’s publication criteria ? Is the manuscript technically sound, and do the data support the conclusions? The manuscript must describe methodologically and ethically rigorous research with conclusions that are appropriately drawn based on the data presented.

Reviewer #1: Yes

Reviewer #2: No

Reviewer #3: Yes

2. Has the statistical analysis been performed appropriately and rigorously?

Reviewer #1: Yes

Reviewer #2: No

Reviewer #3: Yes

3. Have the authors made all data underlying the findings in their manuscript fully available (please refer to the Data Availability Statement at the start of the manuscript PDF file)?

Reviewer #1: Yes

Reviewer #2: Yes

Reviewer #3: Yes

4. Is the manuscript presented in an intelligible fashion and written in standard English?

Reviewer #1: Yes

Reviewer #2: No

Reviewer #3: Yes

5. Review Comments to the Author

Reviewer #1: I have reviewed the article: PDIG-D-24-0522 ‘Exploring perspectives on digital smoking cessation just-in-time adaptive interventions: A focus group study with adult smokers and smoking cessation professionals’

This is an interesting report and an important area of research. Nevertheless I have a few comments that I think may improve the manuscript.

On page 9, line 199 you state that the topic guides were informally piloted. It would be useful to provide additional detail about the pilot structure in the text and if any changes were made to the topic guide as a result.

Participants in general reported both a low level of smoking and low motivation to quit- this is a limitation as the insights of this group may be quite different from a group that contains heavy smokers or those with a greater motivation to quit. I would recommend including this as a limitation in the discussion.

I understand why smoking cessation professionals’ data and that of the smokers were analysed together, however as they are such different groups with different motivations and experiences, at times I find that the two don’t necessarily fit together all that clearly. However I appreciate it may be impractical to have two separate analyses and will leave that to the research team to decide.

Reviewer #2: I have thoroughly reviewed the manuscript. In my assessment, the work does not currently demonstrate a high degree of innovation. However, I have included detailed recommendations aimed at enhancing the manuscript's overall contribution. I hope these suggestions prove helpful.

Abstract

1.abstract mentions the Theoretical Domains Framework (TDF) and Technology Acceptance Model (TAM). Expand slightly on how these frameworks guided the analysis. Were they primarily used for deductive coding? Give a one-sentence indication of the key constructs from these frameworks that emerged as relevant.

2.Saying "four major themes... were identified" isn't as impactful as it could be. Indicate, even roughly, the prevalence or relative importance of these themes. Did one theme emerge far more strongly than others?

3.current concluding sentence ("JITAIs will need to balance aspects...") is rather generic. Sharpen the specific implications of your findings for JITAI design. What is the novel takeaway regarding this balance?

4.author summary uses the pronoun "we" without defining who "we" are.

Introduction

1.While the introduction establishes the importance of smoking cessation and the potential of JITAIs, more could be done to highlight the specific gaps in the current JITAI literature that this study addresses. What are the limitations of existing JITAIs that this research seeks to overcome?

2.rationale for including both smokers and smoking cessation professionals is good, but it could be further strengthened. Expand on the unique insights each group brings to the table. What specific knowledge or experience does each group possess that the other lacks?

3.Make the connection between the research questions and the practical considerations of JITAI design more explicit. How will answering these questions directly inform decisions about intervention features, content, and delivery mechanisms?

4.While the study recognizes the value of autonomy and personalization, it could expand on how these features can be used to address health inequities among diverse populations.

Methods

1.manuscript mentions a pragmatist epistemology, but doesn't explain what that means in the context of this study. How did this epistemological stance influence the research design, data collection, or analysis?

2.While the rationale for the number of focus groups and participants is provided, strengthen the argument for data saturation. What specific criteria were used to determine when saturation was reached?

3.While the topic guides are included as supporting information, provide more detail within the main text about the key areas explored in each guide. What specific prompts or probes were used to elicit participant perspectives on JITAI features, content, and integration with healthcare systems?

4.Expand on the process of developing the codebook. What specific steps were taken to ensure inter-coder reliability and consistency in coding?

Results

• While the results section presents demographic data on participants, integrate this information more effectively into the qualitative findings. How did participant characteristics relate to their perspectives on JITAIs?

• While the results section includes quotations, increase the frequency and diversity of participant voices. Select quotations that are particularly compelling, insightful, or representative of different perspectives.

•manuscript mentions that perspectives on user engagement and privacy diverged between stakeholder groups. Quantify the extent of these differences and provide more detailed examples of these contrasting viewpoints.

•demographic data is described but does not consider how characteristics such as "educational attainment" affect views on JITAIs.

Discussion

1.discussion compares the findings to existing literature, but it could be more nuanced and critical. What are the key points of convergence and divergence between this study's findings and previous research? Are there any findings that contradict or challenge existing assumptions?

2.discussion identifies a tension between convenience and disruption, but it doesn't fully explore the implications of this finding. How can JITAIs effectively balance these competing demands? What specific design strategies can be used to achieve this balance?

3.limitations section could be more specific about the potential impact of the study's limitations on the generalizability of the findings. How might the findings differ if the study had included a more diverse sample?

4."research team and reflexivity" section can be mentioned in the discussion to explain how the researchers' academic backgrounds may influence the way data was collected and interpreted.

5.conclusion should provide a clear and compelling call to action for researchers and practitioners. What specific steps should they take to translate the findings of this study into practice?

6.conclusion should clearly articulate the unique contribution of this study to the existing literature. What new knowledge or insights does this study offer that were not previously available?

7.conclusion should consider the long-term implications of this research for the field of smoking cessation. How might JITAIs transform the way that smoking cessation support is delivered in the future?

Reviewer #3: About the paperwork

The paper titled "Exploring Perspectives on Digital Smoking Cessation Just-in-Time Adaptive Interventions: A Focus Group Study with Adult Smokers and Smoking Cessation Professionals" explores the views of adult smokers and smoking cessation professionals on just-in-time adaptive interventions (JITAIs) for smoking cessation. The study conducted focus groups with UK-based adult smokers and smoking cessation professionals between January and June 2024. The focus group discussions covered various topics, including the integration of JITAIs into users' lives, preferred content and features, and data privacy concerns.

Four major themes and 16 subordinate themes were identified through analysis:

1)Smoking Cessation Process: The non-linear and challenging journey of smoking cessation where JITAIs should provide consistent support.

2)JITAI characteristics: preferences for highly personalized interventions that offer both immediate support and in-depth content.

3)Perceived Value of the JITAI: The perceived usefulness and ease of use of a JITAI, with an emphasis on balancing convenience and disruption to facilitate behavior change.

4)Relationship with the JITAI: The desired supportive and non-judgmental relationship with a JITAI, promoting user autonomy.

In conclusion

The study highlights the need for JITAIs to be convenient, highly personalized, supportive, and non-judgmental while addressing privacy and data protection concerns. Smoking cessation professionals stressed the importance of privacy, while smokers had mixed opinions on this topic.

General comment

This paper provides valuable insights into the perspectives of both smokers and smoking cessation professionals on digital interventions for smoking cessation. The thorough analysis and identification of key themes offer a comprehensive understanding of user needs and preferences. The authors have successfully highlighted the importance of balancing convenience with effective disruption, personalizing interventions, and addressing privacy concerns. This study's findings have the potential to guide the development of more effective and user-friendly JITAIs, ultimately contributing to improved smoking cessation outcomes. Well done on such a well-structured and insightful research paper!

6. PLOS authors have the option to publish the peer review history of their article (what does this mean? ). If published, this will include your full peer review and any attached files.

**Do you want your identity to be public for this peer review?** For information about this choice, including consent withdrawal, please see our Privacy Policy .

Reviewer #1: No

Reviewer #2: No

Reviewer #3: No

---

## [Decision Letter · Decision Letter 1]

1 Apr 2025

Exploring perspectives on digital smoking cessation just-in-time adaptive interventions: A focus group study with adult smokers and smoking cessation professionals

PDIG-D-24-00522R1

Dear Ms Leppin,

We are pleased to inform you that your manuscript 'Exploring perspectives on digital smoking cessation just-in-time adaptive interventions: A focus group study with adult smokers and smoking cessation professionals' has been provisionally accepted for publication in PLOS Digital Health.

Best regards,

Haleh Ayatollahi

Section Editor

PLOS Digital Health

**Additional Editor Comments (if provided):**

**Reviewer Comments (if any, and for reference):**

Reviewer's Responses to Questions

**Comments to the Author**

1. If the authors have adequately addressed your comments raised in a previous round of review and you feel that this manuscript is now acceptable for publication, you may indicate that here to bypass the “Comments to the Author” section, enter your conflict of interest statement in the “Confidential to Editor” section, and submit your "Accept" recommendation.

Reviewer #1: All comments have been addressed

Reviewer #2: All comments have been addressed

2. Does this manuscript meet PLOS Digital Health’s publication criteria ? Is the manuscript technically sound, and do the data support the conclusions? The manuscript must describe methodologically and ethically rigorous research with conclusions that are appropriately drawn based on the data presented.

Reviewer #1: Yes

Reviewer #2: (No Response)

3. Has the statistical analysis been performed appropriately and rigorously?

Reviewer #1: Yes

Reviewer #2: (No Response)

4. Have the authors made all data underlying the findings in their manuscript fully available (please refer to the Data Availability Statement at the start of the manuscript PDF file)?

Reviewer #1: Yes

Reviewer #2: (No Response)

5. Is the manuscript presented in an intelligible fashion and written in standard English?

PLOS Digital Health does not copyedit accepted manuscripts, so the language in submitted articles must be clear, correct, and unambiguous. Any typographical or grammatical errors should be corrected at revision, so please note any specific errors here.

Reviewer #1: Yes

Reviewer #2: (No Response)

6. Review Comments to the Author

Please use the space provided to explain your answers to the questions above. You may also include additional comments for the author, including concerns about dual publication, research ethics, or publication ethics. (Please upload your review as an attachment if it exceeds 20,000 characters)

Reviewer #1: (No Response)

Reviewer #2: After reviewing the changes made by the authors, I can see that they have attended to the revisions appropriately and the manuscript seems acceptable to be published within this form.

7. PLOS authors have the option to publish the peer review history of their article (what does this mean? ). If published, this will include your full peer review and any attached files.

**Do you want your identity to be public for this peer review?** For information about this choice, including consent withdrawal, please see our Privacy Policy .

Reviewer #1: No

Reviewer #2: **Yes: ** Roghieh Nooripour
